# The Intertwined Role of 8-oxodG and G4 in Transcription Regulation

**DOI:** 10.3390/ijms24032031

**Published:** 2023-01-19

**Authors:** Francesca Gorini, Susanna Ambrosio, Luigi Lania, Barbara Majello, Stefano Amente

**Affiliations:** 1Department of Molecular Medicine and Medical Biotechnologies, University of Naples Federico II, 80131 Naples, Italy; 2Department of Biology, University of Naples Federico II, 80138 Naples, Italy

**Keywords:** 8-oxodG, G4, PQS, BER, transcription regulation, DNA damage

## Abstract

The guanine base in nucleic acids is, among the other bases, the most susceptible to being converted into 8-Oxo-7,8-dihydro-2′-deoxyguanosine (8-oxodG) when exposed to reactive oxygen species. In double-helix DNA, 8-oxodG can pair with adenine; hence, it may cause a G > T (C > A) mutation; it is frequently referred to as a form of DNA damage and promptly corrected by DNA repair mechanisms. Moreover, 8-oxodG has recently been redefined as an epigenetic factor that impacts transcriptional regulatory elements and other epigenetic modifications. It has been proposed that 8-oxodG exerts epigenetic control through interplay with the G-quadruplex (G4), a non-canonical DNA structure, in transcription regulatory regions. In this review, we focused on the epigenetic roles of 8-oxodG and the G4 and explored their interplay at the genomic level.

## 1. Introduction

8-Oxo-7,8-dihydro-2′-deoxyguanosine (8-oxodG), the major product of oxidative DNA damages, has been proposed as an epigenetic marker in gene regulation, and it has been associated with genome instability, cancer, neurological diseases and aging [1].

Since guanine has the lowest oxidation potential of all the DNA bases [2,3], it is largely damaged by reactive oxygen species (ROS) (Figure 1a). Thus, 8-oxodG represents the most prevalent oxidized form of guanine in the genome [4,5,6]. 8-oxodG can lead to the incorrect incorporation of dA at the improper place during DNA replication, which may cause a dG-to-dT transversion mutation. In human cells, OGG1 (8-Oxoguanine DNA glycosylase) and MUTYH (MutY DNA glycosylase) glycosylases clear 8-oxodGs and the misincorporated dAs to prevent this harmful effect. Then, the DNA glycosylases leave apurinic (AP) sites that are then handled by the other proteins of the base excision repair pathway (BER) [1].

DNA glycosylases are categorized into two classes—monofunctional (e.g., MUTYH) and bifunctional (e.g., OGG1)—based on their AP lyase activity. Monofunctional DNA glycosylases require separate AP endonucleases (APE1 and APE2), whereas bifunctional endonucleases are sufficient to create a single-strand break (SSB) by the cleavage of the AP sugar-phosphate backbone. To repair the damage, the resulting gap in the DNA is filled and rejoined by either replacing the AP site with an appropriate single-nucleotide match (a short-patch BER) or synthesizing a few long matches (a stretch of 2–10 nucleotides, a long-patch BER) [1].

While low levels of 8-oxodG make it easier to maintain telomeres’ integrity, chronic 8-oxodG lesions hinder telomere replication. Thus, 8-oxodG has a role in telomerase stability [7]. Moreover, the accumulation of 8-oxodG at specific DNA regions has also been associated with double-strand break (DSB) formation and genome instability [1]. For all the mentioned reasons, 8-oxodG is frequently referred to as a form of DNA damage. In addition, mounting evidence recently demonstrated that 8-oxodG participates in the epigenetic control of transcription [1,8,9,10].

Intriguingly, several lines of evidence support interplay between guanine oxidation and the G-quadruplex (G4) folding in transcription regulatory regions [11,12,13,14,15,16,17].

The G4 is a noncanonical DNA secondary structure which can fold at specific G-rich regions and compete with the canonical B-form double-stranded DNA. The G-rich region containing at least four stretches of three Gs (G runs) which can fold in the G4 is referred to as the Potential Quadruplex Sequence (PQS). At these regions, four Gs can be held together through Hoogsteen hydrogen bonding and are stabilized by alkali cations such as K^+^, forming a square planar structure named G-tetrad. Three G-tetrads can stack together forming a G4 structure (Figure 1b) [18,19]. A G4 can have three topologies, anti-parallel, parallel and hybrid structures, depending on the relative orientation of the DNA strand within the structure (Figure 1c) [20,21]. PQSs and G4s have been found enriched at gene regulatory regions such as promoters/enhancers and at telomeres [22,23]. Nonetheless, the biological role of G4s is still the object of extensive studies and needs to be better characterized.

Many studies report their involvement in physiological molecular mechanisms such as transcription, DNA replication, DNA repair, telomere maintenance, and genetic and epigenetic instability [24,25,26,27,28,29].

## 2. PQS/G4 in Transcription Regulation

G4s can function as binding hubs for many transcription factors (e.g., MAZ and SP1) to promote transcription (Figure 2a) [30,31,32,33,34,35]. Moreover, promoters of many oncogenes and other cancer-associated genes (e.g., *MYC*, *KRAS*) have the PQS and G4 which appear to regulate their expression [36]. G4s have been initially described to function as inhibitors of transcription. Experiments performed in cells treated with a TMPyP4 (a G4 stabilizing ligand) have revealed a transcription inhibition of the *MYC* gene, thus suggesting that the stabilization of the G4 structure within the *MYC* promoter may cause a roadblock to its transcription (Figure 2b) [37]. However, many other studies showed that the role of G4 structure in transcription is more complex as they can also enhance transcription by altering the epigenetic marks and chromatin architecture. This aspect is addressed and discussed below in this review.

Recently, several NGS-based methods have been developed to map the position of G4s or to predict the PQS [36,38,39,40,41]; these methodologies helped the investigation of the role of the G4 in physiological and pathological processes. Via G4-Seq analysis, G4s have been described to map G4s at promoters and 5′UTR regions of highly transcribed genes and among cancer-related genes and somatic-copy-number-variation-containing regions [36]. The ChIP-Seq-based technique, named G4-ChIP-Seq and G4P ChIP-Seq, also greatly helped the investigation into G4′s role in transcription [36,41]. The G4P ChIP-Seq technique uses a G4 probe (an artificial protein with high affinity and specificity for G4) to pull down and enrich G4-containing DNA fragments. According to the other methods, G4P-ChIP-Seq found an accumulation of G4s at promoter regions [41]. Finally, a G4-CUT&TAG method, also developed to identify G4s with higher sensitivity starting from a small cell number, found the co-occurrence of G4s with single-stranded DNA and R loops at promoters and enhancers [42].

All these new methodological approaches will be instrumental to better dissect the regulatory role of G4 in gene expression.

## 3. G4 as a Pharmacological Target

According to various evidence, changes in G4 formation/stability have been shown to affect telomerase activity, transcription efficiency, DNA replication arrest, and affect genomic stability [43,44,45,46]. Thus, G4 ligands, which regulate the production of G4 structures or stabilize G4 structures, have been developed with the aim to produce anti-tumor therapeutic drugs by preventing cellular replication or oncogene expression through G4 formation/stabilization [47,48].

To date, different G4 ligands have been discovered, differing in selectivity, binding surface, and cell permeability [49]. Most of them, including MM41 [50], Telomestatin [51], BRACO19 [52], TMPyP4 [53], RHPS4 [54], and Pyridostatin (PDS) [55], bind G4 structures more frequently than duplex DNA. Most of these studies have only been performed in vitro, and it remains to be determined whether or how selectively these G4 ligands function in living cells. Additionally, various evidence suggests that at least some G4 ligands (such as BRACO19) also bind to other non-canonical DNA structures such as the i-motif, raising the possibility that some of the effects of G4 ligands in living cells may not result exclusively from G4 stabilization [56,57]. Nevertheless, some of these ligands are promising drugs for therapeutic purposes, although their application in clinics has not yet been approved [58]. It remains necessary to improve their selectivity since the majority of G4 ligands target different G4s across the genome.

In the recent literature, a great deal of work regarding the search for G4 ligands by selectively targeting specific G4 structures can be found, with the aim to use G4 ligands as precise anti-tumoral drugs with few adverse side effects [59,60].

## 4. 8-oxodG in Transcription Regulation

Evidence based on recent studies demonstrates that 8-oxodG and its repair intermediates have epigenetic-like features, mediating the up- or downregulation of gene expression [61,62,63,64,65,66,67,68,69,70,71,72,73,74,75,76,77], and three different models have been proposed.

In one model, the recruitment of the co-transcription factor LSD1, a histone demethylase enzyme, is associated with elevated amounts of 8-oxodG in the binding areas of specific transcription factors (TFs) [61,62,63,64,65,66,67,68]. Based on this model, specific TFs (estrogen receptor, MYC, androgen receptor, and TGF-β1) recruit the LSD1 enzyme, which in turn promotes transcription activation by enhancing 8-oxodG formation as a result of its enzymatic activity. Repairing 8-oxodG causes the formation of a DNA nick which facilitates the entrance of the transcription machinery, promoting transcription initiation (Figure 3a).

In an alternative proposed model, the binding of OGG1 to 8-oxodG in the promoter regions of NF-kB target genes results in the induction of gene expression [69,70]. Based on this model, the recruitment of TFs, such as NF-kB/RelA and Sp1, and the consequent assembling of the transcriptional machinery are facilitated by the binding of enzymatically inactive OGG1 to the 8-oxodG-containing promoter [69,70] (Figure 3b).

More recently, it has been shown that the synthetic alteration of 8-oxodG in promoter regions of specific genes, such as *VEGF* and *NTHL1*, activated their transcription [71]. For these genes, the “spare tire” model has been proposed, in which OGG1 and APE1 promote gene expression [71,72] (Figure 3c). This model is better discussed in the next section. A revision of this model has also been recently proposed in [73].

The oxidation of 8-oxodG in PQSs could also repress transcription, as demonstrated for the *RAD17* and *NEIL3* genes, through topological alterations of the G-quadruplex [74,75]. Overall, even if the role of 8-oxodG in repairing intermediates (e.g., AP sites and SSBs) in gene expression remains to be better investigated, the role of 8-oxodG in the epigenetic modulation of gene transcription appears clear.

Recently, different NGS-based methodologies and the development of bioinformatic tools greatly helped the identification of the precise location of 8-oxodG in yeast, mouse, and human genomes. All these methodologies rely on enrichment by the pull down of oxidized genomic DNA followed by sequencing [11,12,16,76,77]. These genome-wide-based approaches revealed a non-random distribution of 8-oxodG along the genome with its accumulation at G-rich regulatory regions, such as promoters and enhancers. As has been effectively described in [1], the non-random distribution of 8-oxodG can be due to either the specific local formation of ROS (Figure 3a) or an imbalance between the formation and repair of the 8-oxodG (Figure 3b,c). This supports the idea that 8-oxodG formation and/or accumulation, as part of a transcription program controlled by specific TFs [61,62,63,64,65,66,67,68,69,70,71,72,73,74,75], has an epigenetic-like role in transcription regulation.

## 5. The Interplay of 8-oxodG and G4 in Transcription

Given that 8-oxodG and the G4 colocalize at G-rich regulatory regions, it is not surprising that these two structures are often intertwined in transcription regulation and chromatin organization and perhaps are also interdependent and cross functional. While 8-oxodG has a weak influence on the stability of the double helix form of the DNA, it can destabilize the G4 structure. Indeed, the oxygen on the C8 of 8-oxodG causes the protonation of its N7, which results in a hydrogen-bond donor, instead of acceptor, and this prevents the formation of Hoogsteen base pairs required for the stabilization of the G4 structure [78]. In contrast to 8-oxodG, AP sites resulting from 8-oxodG removal via OGG1 in the PQS favor G4 formation at the expense of the double-helix DNA structure [63,78,79]. Thus, 8-oxodG and its associated BER proteins can change the balance between G4 and the double helix of the DNA.

Using the luciferase reporter gene under the control of PQS-containing VEGF and NTHL1 promoters, it has been shown that the plasmid containing an 8-oxodG in the PQS promoters determined an increase in luciferase expression when compared with the control plasmid without 8-oxodG [71]. This increased expression was associated with the removal of 8-oxodG via OGG1, AP site formation, APE1 recruitment, and G4 formation [71]. However, it is worth noting that, given that ~40% of all PQSs have five G runs, promoters used in this work are characterized by the presence of a fifth extra run of guanines, and the “spare tire” model has been proposed (Figure 3c). For this model, the oxidation of a G determines the recruitment of OGG1 in the damaged G run of the PQS. OGG1, excising 8-oxodG, generates an AP site, and the relative AP-containing region is then extruded into a loop. Subsequently, the replacement of this extruded damaged G run with the fifth extra G run stabilizes an alternative G4 structure formation. Then, the APE1 binds to the AP site into the extruded loop and, independently of its endonuclease activity, promotes the recruitment transcription factors (e.g., HIF1α, STAT3, and CBP/P300) for transcription activation (Figure 3c) [63,80,81]. However, although it is not clear whether 8-oxodG’s repair intermediates in the G4s or in its associated extruded loop are efficiently repaired, the role of 8-oxodG and APE1 in the activation of the report luciferase gene is evident. Similar results have been obtained when a plasmid bearing the luciferase gene, under the control of the PQS-containing promoter of the human *NEIL3* gene, has been transfected in human glioblastoma cells induced with the cytokine TNFα to stimulate physiological 8-oxodG formation in the *NEIL3* PQS-containing promoter [13]. The use of plasmid systems, which cannot accurately reflect the function of G4s in a chromatin environment, is an important caveat of these investigations.

Additionally, a different methodology capable of identifying naturally occurring G4s, utilizing G4 ChIP-Seq, revealed a high enrichment of APE1 at G4 locations in human cells [82,83]. This study further established a mechanism that associates G4 oxidation with increased transcriptional activity in vivo by demonstrating that the binding of APE1 to G4s stabilizes the G4 structure and lengthens its residence time at DNA damage sites [82,83].

Moreover, a recent report demonstrated that oxidized quadruplexes can be bound by and improve the activity of poly ADP-ribose polymerase-1 (PARP-1), another crucial component of the BER pathway. As APE1, PARP1 appears to respond to the formation of the looped G4 structure which results from DNA damage [84,85].

Finally, a study conducted in pancreatic cancer cells revealed that 8-oxodG formation may facilitate transcription via the recruitment of the transcription factor proteins MAZ and hnRNP A1 to the G4-rich *KRAS* promoter [86]. All these studies suggest that the 8-oxodG formation is not a negative event as its repair (through the BER pathway) can stimulate G4 formation for transcription activation.

While the effect of 8-oxodG on G4 has been well documented, the impact of G4 on 8-oxodG has not. We know that PQSs and G4s could be hot spots of 8-oxodG because they contain at least four G runs, and the 5’ G of a G run is more easily oxidized [81,87]. Indeed, in vitro experiments showed that the formation of G4 alters the frequency and distribution of G oxidation products along a PQS, compared to dsDNA [88]. However, this work was performed in vitro and under specific reaction conditions; hence, it may fail to capture the complex interactions between G4s and 8-oxodG within cells [88]. Moreover, the G4 structure is expected to influence the repair of G oxidation because OGG1 can only recognize 8-oxodG in double-helix DNA and not in single-strand DNA or in the G4 structure.

Key questions are still open: are the G4 hot spots of 8-oxodG also in a chromatin context? Does 8-oxodG accumulation, following oxidative stress, influence the steady-state level and genome-wide distribution of G4 in vivo? Does G4 stabilization or destabilization, following G4 binder or G4 helicase (which unwind G4 secondary structures) overexpression, determine a genome-wide alteration of the 8-oxodG level and distribution? In other words, does a relationship between 8-oxodG and G4 exist in the chromatin context for transcription regulation?

As discussed above, NGS-based techniques have recently been developed to detect the genomic distribution of G4s and 8-oxodG and only marginally help to answer those questions. The studies based on the use of such techniques revealed 8-oxodG and G4 were enriched in promoter regions, further supporting their epigenetic role in gene transcription [11,12,15,16,76,89]. However, these studies also showed that, whilst sets of oxidized promoters are enriched for the presence of G4s, only a few G4-containing promoters are oxidized. This suggests that G4s are not hot spots of 8-oxodG in vivo; hence, a more intricated crosstalk between 8-oxodG and G4s for transcription regulation may exist.

## 6. The Potential Interplay of 8-oxodG and G4 in Chromatin Organization

Recently, studies revealed that both 8-oxodG and G4s may be involved in the regulation of long-range chromatin interactions. In one of these studies, it has been found that G4s are abundant at loop boundaries [90]. Moreover, it has also been discovered that the presence of G4s at loop boundaries increases the binding of the transcription factor, the stability of DNA looping, and then long-distance DNA interactions [90]. This is also supported by data showing that PQSs are enriched at the CTCF motif [90]. Hence, it has been speculated that G4s may cooperate with CTCF proteins to impede cohesion progression and define loop boundaries of promoter–enhancer interactions.

Intriguingly, 8-oxodG has been also found to accumulate at enhancer regions [89]. Specific transcription factors involved in the inflammatory response pathways are associated with 8-oxodG-enriched enhancers [89]. Moreover, it has been discovered that oxidized enhancers are physically associated with oxidized promoters within oxidized specific CTCF-mediated chromatin loops [89]. In addition, 8-oxodG-enriched enhancer–promoter pairs have been found at the boundaries of specific RNAPII-mediated chromatin loops [89], implying that the oxidation of such enhancers and target gene promoters occur concurrently during the transcription process. This supports a model in which the accumulation of 8-oxodG at enhancer–promoter pair contacts is likely due to transcription factors that, once recruited at enhancer and promoter regions, establish crosstalk, promoting chromatin looping.

Overall, the combination of specific NGS-based approaches for G4 or 8-oxodG genome mapping to methods capable of capturing the 3D chromatin organization adds new mechanistic insights into the role of G4 and 8-oxodG in chromatin looping and transcription regulation. However, the crosstalk between 8-oxodG and G4 structures for chromatin looping has not yet been formally demonstrated in vivo.

## 7. 8-oxodG and BER’s Proteins as a Pharmacological Target

The formation and processing of 8-oxodG within promoters and 5′ untranslated regions via BER has been demonstrated to modulate transcriptional activity of genes (*PCNA, KRAS, MYC,* and *VEGF*) and transcription factors (NF-kB), involved in cell proliferation and cancer initiation or progression [91], suggesting a role for BER intermediates in the development of various tumors. While the use of PARP inhibitors to target the BER pathway is already used to treat some forms of cancer [92], alternative BER targeting strategies have surprisingly gained little attention. The pharmacological targeting of oxidative BER proteins in cancer therapies has resulted in the identification of OGG1 as an anti-tumor cancer target. Mainly, Hanna et al. developed TH5487, an efficient drug for OGG1 inhibition [93]. Treatment with TH5487 causes an accumulation of 8-oxodG lesions in the whole genome. This compound disrupts OGG1’s chromatin binding, causing OGG1’s impaired recruitment to DNA damage areas. Moreover, TH5487 interferes with OGG1’s incision activity, and consequently, DNA double-strand breaks are reduced in cells under oxidative stress. Furthermore, it has been shown that the same compound inhibits BER initiation at telomeres under oxidative stress conditions, leading to the accumulation of oxidized bases, association with telomere losses, the development of micronuclei, and mild proliferation problems [94]. Another OGG1 inhibitor, SU0268, has been shown to increase genomic 8-oxodG levels in cancer cells, while it had moderate cytotoxicity and good permeability in normal cells [95]. Finally, Methoxyamine (MX or TRC102), which inhibits the BER pathway by binding and modifying AP sites, has completed phase I clinical trials as a chemosensitizing agent in association with the antifolate pemetrexed in solid tumors [96,97].

Thus, further studies are proceeding to produce novel inhibitors of BER’s proteins which have been accurately reviewed in previous studies [98]. The development of these new compounds will be helpful to verify several hypotheses regarding the function of 8-oxodG in transcription regulation under various physiological and pathological conditions.

## 8. Perspectives

In this review, we have explored the relationship between 8-oxodG and G4. Several data derived from experiments performed using both specific single PQS/G4-containing gene promoters cloned in the gene reporter system and new NGS-based methods have been analyzed. However, the mechanisms and consequences of 8-oxodG and G4 crosstalk remain largely underexplored.

The investigation of how 8-oxodG and G4s cooperate in vivo to control transcription and genome integrity is made possible by mapping G4s and 8-oxodGs at the genomic level. However, the existing research tools have number of limitations. As already mentioned, each sequencing technique recently developed to map genomic distribution of 8-oxodG and G4 in has its drawbacks. The development of 8-oxodG/G4 genome mapping methods using long-read sequencing technologies may help to overcome such limitations. In addition, recently established methods for 8-oxodG genome mapping exclude 8-oxodG oxidation products, such as Gh and Sp, and 8-oxodG repair intermediates. Indeed, while sequencing Gh or Sp is still not possible, there are a few approaches for mapping AP-sites in the genome. Hence, clarifying the mechanism of 8-oxodG-G4 crosstalk requires integrating information from various 8-oxodG-derivates. Finally, the pharmacological and/or genetic targeting of 8-oxodG-processing BER proteins and of G4 structures have been demonstrated as being capable of modifying the levels of both 8-oxodG and G4. Experiments of 8-oxodG/G4 genome mapping performed in cells treated with such drugs, or in genetically manipulated cells to affect BER function or G4 helicases, combined with other omics capable of capturing different levels of the transcription regulation, may help to establish the crosstalk of 8-oxodG and G4 in gene transcription. This may be crucial for examining transcriptional dysregulation in diseases such as cancer, neurodegeneration, and aging.

## Figures and Tables

**Figure 1 ijms-24-02031-f001:**
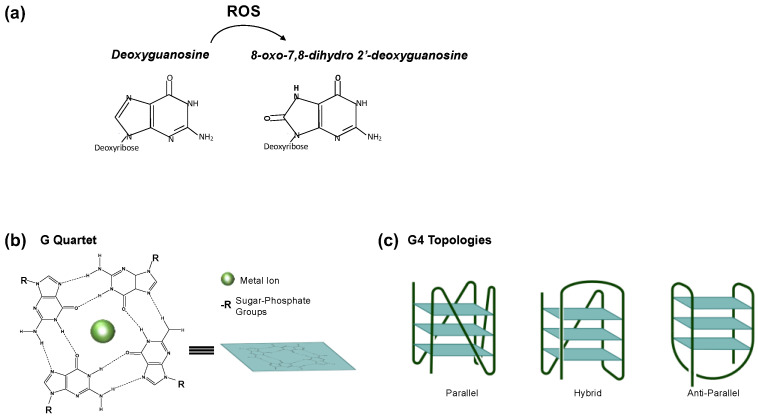
(**a**) 8-oxo-7,8-dihydro-deoxyguanosine (8-oxodG) chemical structure. 8-oxodG is produced when ROS oxidizes 2′-deoxyguanosine. (**b**) Schematic representation of G-quadruplex structures. (**c**) Depending on the relative orientation of the DNA strand within the structure, specific G4 topologies can be created, including anti-parallel, parallel and hybrid structures.

**Figure 2 ijms-24-02031-f002:**
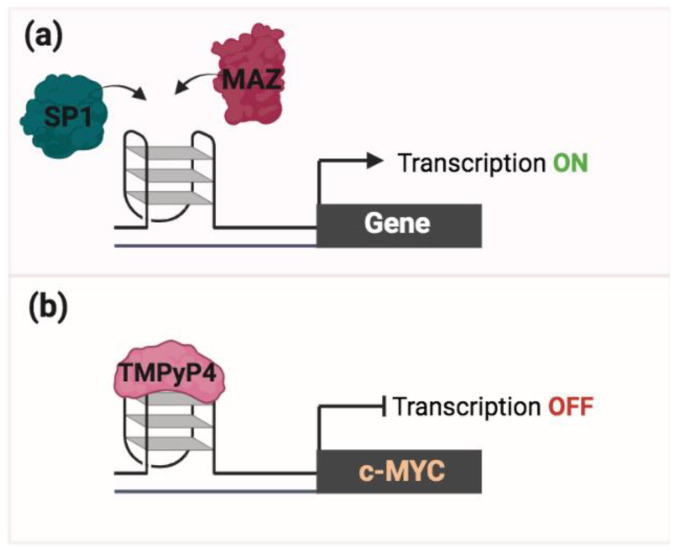
Epigenetic role of G4. (**a**) G4s in the promoter regions of specific genes promote the recruitment of specific TF (SP1 and MAZ) to regulate their expression. (**b**) Stabilization of the G4 structure by TMPyP4, within the *MYC* promoter, causes a roadblock to MYC transcription.

**Figure 3 ijms-24-02031-f003:**
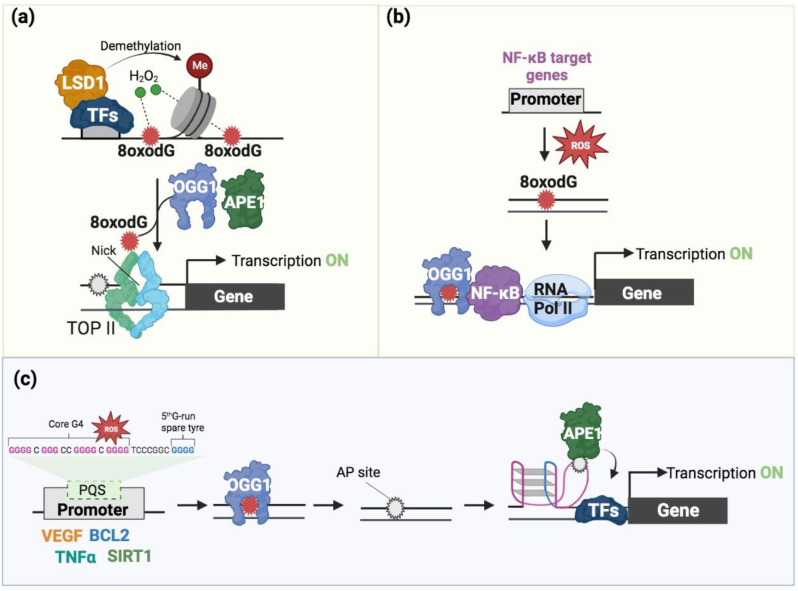
Epigenetic role of 8-oxodG. (**a**) A specific TF recruits LSD1 demethylases to its target genes. Local ROS (H2O2), produced by LSD1 during the histone demethylation reaction, results in the formation of 8-oxodG. 8-oxodG is then converted in an AP site via OGG1 and subsequently in a nick via APE1. The nick allows the entry of the Topoisomerase II and of the RNA polymerase II machinery to activate the transcription of downstream genes. (**b**) The signaling pathway activated by ligands (such as TNF) produces ROS which induce 8-oxodG formation close to NF-kB binding sites. 8-oxodG promotes the recruitment of the OGG1-NF-kB complex, thus activating the transcription of downstream genes. (**c**) “Spare tire” model. ROS promotes the formation of 8-oxodG (red) in the PQS-containing promoter of specific genes (VEGF, BCL2, etc.); the PQS has a fifth extra run of guanines (indicated in blue) in addition to the typical four G runs (indicated in pink). OGG1 is recruited and promotes the AP site (gray) formation as an intermediate of the 8-oxodG-repair process. The AP-containing region is then extruded into a loop, and the fifth extra G run stabilizes an alternative G4 structure formation. Then, APE1 binds to the AP site into the extruded loop and, independently of its endonuclease activity, promotes the recruitment of various TFs to activate the transcription of downstream genes.

## Data Availability

Not applicable.

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
