# Peer review of "The Intertwined Role of 8-oxodG and G4 in Transcription Regulation"

_ijms, 2023, doi:10.3390/ijms24032031_

Round 1

Reviewer 1 Report

A review of Francesca Gorini and coworkers addresses a very interesting and currently actively researched topic devoted to the interplay between 8-oxodG and G4-structures and their influence on chromatin organization in general and transcriptional activity in particular. The review systematizes results of over 100 papers in this area and may be of interest to a wide range of researchers.However, for its publication, the review requires some clarifications and adjustments.

 Major remarks

 The model describes in lines 126-137 is not completely clear and it should be explained better. In particular, the authors write that “Repairing of 8-oxodG causes the formation of a DNA nick which facilitates the entrance of the transcription machinery promoting transcription initiation.” And then they note that “the recruitment of TFs … and the consequent assembling of the transcriptional machinery are facilitated by the binding of enzymatically inactive OGG1 to the 8-oxodG-contain promoter”. But it is unclear how the binding of enzymatically inactive OGG1 can result in nick formation, and therefore it is unclear what is the driving force for transcription initiation.

 In lines 154-155 the authors write “the G-quadruplex structure, by extending the catalytically inactive APE1's binding to the AP site, promotes the recruitment of specific transcription factors” but it is not completely correct since in the cited works APE1 inhibitors and APE1 knockdown were used, but not catalytically inactive APE1. It was done in order to demonstrate that gene activation occurs due to APE1 binding and it does not require the phosphodiesterase activity of APE1. Once again, the authors should describe better the model proposed for participation of BER enzymes in the activation of transcription from promoters containing 8-oxodG and G4-structures. Moreover, given that this model is related to the interaction between 8-oxodG repair and G-quadruplex formation, the reviewer proposes to combine section 4 and section 6, especially since in section 6 the authors again return to work 71, which they cite in section 4. Combining these sections would make them more understandable, especially if the authors give a figure in which they show how the oxidation of G and the AP-site formation contributes to the formation of G-quadruplex. Obviously, it is also necessary to explain that during the oxidation of G, the electron density is redistributed, which makes it impossible for the formation of Hoogsteen pairs and, due to this, destabilizes the G4.

 Miner remarks

Lines 77-79: in the sentence ‘its role has been also found to enhance transcription” the word “role” would be better replaced by the word "presence" or "formation".

 Lines 123-125: the sentence needs some references.

 Lines 195-196: it is not clear what processes the authors talk about in the phrase “Given that 8-oxodG and G4 colocalize at G-rich regulatory regions, it is not surprising that these two processes are often intertwined in transcription regulation…” since 8-oxodG and G4 are structures and not processes.

 Lines 243-245: this statement needs references.

Author Response

Response to reviewers’ comments

We thank the editor and the reviewers for their time and effort in reviewing our manuscript. Please find below a point-by-point response to each of the reviewers’ suggestions and concerns.

Review Report (Reviewer 1)

A review of Francesca Gorini and coworkers addresses a very interesting and currently actively researched topic devoted to the interplay between 8-oxodG and G4-structures and their influence on chromatin organization in general and transcriptional activity in particular. The review systematizes results of over 100 papers in this area and may be of interest to a wide range of researchers. However, for its publication, the review requires some clarifications and adjustments.

Major remarks

The model describes in lines 126-137 is not completely clear and it should be explained better. In particular, the authors write that “Repairing of 8-oxodG causes the formation of a DNA nick which facilitates the entrance of the transcription machinery promoting transcription initiation.” And then they note that “the recruitment of TFs ... and the consequent assembling of the transcriptional machinery are facilitated by the binding of enzymatically inactive OGG1 to the 8-oxodG-contain promoter”. But it is unclear how the binding of enzymatically inactive OGG1 can result in nick formation, and therefore it is unclear what is the driving force for transcription initiation.

We thank the reviewer for this comment. We amended the text according to his suggestions. We hope that in the revised version the three-emerging model through which 8-oxodG exerts its epigenetic function is evident.

In lines 154-155 the authors write “the G-quadruplex structure, by extending the catalytically inactive APE1's binding to the AP site, promotes the recruitment of specific transcription factors” but it is not completely correct since in the cited works APE1 inhibitors and APE1 knockdown were used, but not catalytically inactive APE1. It was done in order to demonstrate that gene activation occurs due to APE1 binding and it does not require the phosphodiesterase activity of APE1.

We thank the reviewer for this comment. We amended the text according to his suggestions.

Once again, the authors should describe better the model proposed for participation of BER enzymes in the activation of transcription from promoters containing 8-oxodG and G4-structures. Moreover, given that this model is related to the interaction between 8-oxodG repair and G-quadruplex formation, the reviewer proposes to combine section 4 and section 6, especially since in section 6 the authors again return to work 71, which they cite in section 4. Combining these sections would make them more understandable, especially if the authors give a figure in which they show how the oxidation of G and the AP-site formation contributes to the formation of G-quadruplex.

We thank the reviewer for this comment. We amended the text according to his suggestions and moved the paragraph “8-oxodG and BER’s proteins as a pharmacological target“ to the end of the text, before the “Perspectives”. In this manner, the paragraphs “8-oxodG in transcription regulation” and “The interplay of 8-oxodG and G4 in transcription“ result back-to-back and the information is more fluent.

In addition, we modified the Figure 3c and the text to better describe how the oxidation of G and the AP-site formation contributes to the formation of G-quadruplex.

Obviously, it is also necessary to explain that during the oxidation of G, the electron density is redistributed, which makes it impossible for the formation of Hoogsteen pairs and, due to this, destabilizes the G4.

We thank the reviewer for this comment. We amended the text according to his suggestion.

Miner remarks

Lines 77-79: in the sentence ‘its role has been also found to enhance transcription” the word “role” would be better replaced by the word "presence" or "formation".

Lines 123-125: the sentence needs some references.

Lines 195-196: it is not clear what processes the authors talk about in the phrase “Given that 8-oxodG and G4 colocalize at G-rich regulatory regions, it is not surprising that these two processes are often intertwined in transcription regulation...” since 8-oxodG and G4 are structures and not processes.

Lines 243-245: this statement needs references.

We thank the reviewer for these comments. We amended the text to accommodate all his suggestions.

We agree and appreciate the reviewer’s constructive comments, and we made our best efforts to address all the reviewers’ suggestions. We hope that the revision done will be sufficient to satisfy all the referee’s concerns.

Reviewer 2 Report

This review is generally well written and the figures are informative. At times the English usage is a bit odd, and this could be improved upon, although even as it is written now, it is easily understood. 

The main comment that I would make is that the review doesnt really make it clear how a damage product, that I guess is generated randomly, could be part of a programme to control transcription and chromatin structure. Does this only apply to specific damage related genes, or is it thought to be more widespread. Is the oxo-G accumulation specific to G4 structures or do regions of the genome that  are G rich but dont form G4 structures show the same accumulation of oxidation.  I think that a little more information in these areas would give the review more context for the reader.

Author Response

We thank the editor and the reviewers for their time and effort in reviewing our manuscript. Please find below a point-by-point response to each of the reviewers’ suggestions and concerns.

Review Report (Reviewer 2)

This review is generally well written and the figures are informative. At times the English usage is a bit odd, and this could be improved upon, although even as it is written now, it is easily understood. 

The main comment that I would make is that the review doesnt really make it clear how a damage product, that I guess is generated randomly, could be part of a programme to control transcription and chromatin structure. Does this only apply to specific damage related genes, or is it thought to be more widespread. Is the oxo-G accumulation specific to G4 structures or do regions of the genome that  are G rich but dont form G4 structures show the same accumulation of oxidation.  I think that a little more information in these areas would give the review more context for the reader.

We thank the reviewer for these comments. We amended the text and undelighted that the 8-oxodG is not randomly distributed along the genome and this can be due to either specific local formation of ROS (Figure 3a) or an imbalance between formation and repair of the 8-oxodG (Figure 3b and c). This supports the idea that 8-oxodG formation and/or accumulation, as part of a transcription program controlled by specific has an epigenetic-like role in transcription regulation.

We agree and appreciate the reviewer’s constructive comments, and we made our best efforts to address all the reviewers’ suggestions. We hope that the revision done will be sufficient to satisfy all the referee’s concerns.

Round 2

Reviewer 1 Report

The manuscript has been sufficiently improved and now it can be published in IJMS.